# Electric Vehicle Fast Charging Needs in Cities and along Corridors

**Rick Wolbertus [1,2,]***  **and Robert Van den Hoed [1]**

[1]  Department of Urban Technology, Faculty of Technology, Amsterdam University of Applied Sciences, 1097 DZ Amsterdam, The Netherlands; r.van.den.hoed@hva.nl
[2]  Transport and Logistics Group, Department of Engineering Systems and Services, Faculty of Technology, Policy and Management, Delft University of Technology, 2628 BX Delft, The Netherlands
*   Correspondence: r.wolbertus@hva.nl

**Abstract:** Fast charging is seen as a means to facilitate long-distance driving for electric vehicles (EVs). As a result, roll-out planning generally takes a corridor approach. However, with higher penetration of electric vehicles in urban areas, cities contemplate whether inner-city fast chargers can be an alternative for the growing amount of slow public chargers. For this purpose, more knowledge is required in motives and preferences of users and actual usage patterns of fast chargers. Similarly, with increasing charging speeds of fast chargers and different modes (taxi, car sharing) also switching to electric vehicles, the effect of charging speed should be evaluated as well as preferences amongst different user groups. This research investigates the different intentions and motivations of EV drivers at fast charging stations to see how charging behaviour at such stations differs using both data analysis from charging stations as a survey among EV drivers. Additionally, it estimates the willingness of EV drivers to use fast charging as a substitute for on-street home charging given higher charging speeds. The paper concludes that limited charging speeds imply that EV drivers prefer parking and charging over fast charging but this could change if battery developments allow higher charging speeds.

**Keywords:** electric vehicles; fast charging; public charging; level 2 charging; choice experiment; survey

## 1. Introduction

Fast charging has mainly been considered as a means to accommodate long-distance driving for electric vehicles. Many roll-out strategies, therefore, focus on a corridor approach [1,2]. However, with increasing fast charging rates (from 50 kw to 350 kw), the time needed for fast charging approaches fossil fuel refuelling times [3]. Such rates may possibly reduce the need for slow (level 2) charging stations especially in urban areas. These slower charging stations have a significant impact on public space and their business case is difficult due to lower charging speeds and, therefore, lower charge volumes. This provides a need for a better understanding under which circumstances fast charging may provide a realistic alternative to level 2 public charging. Additionally, modalities such as taxis [4,5] and car sharing [6] are also switching to electric vehicles. They will make use of the same fast charging infrastructure. Such modalities have different usage patterns and also, therefore, different needs for fast charging.

Policy makers and charging point operators are struggling with the roll-out of fast charging stations as they are unaware of the intentions of those using fast charging infrastructure. Usage patterns of fast charging stations and level 2 charging stations differ, suggesting different intentions of electric vehicle (EV) drivers for recharging. For slow level 2 charging infrastructures, the intentions can be derived from charging patterns [7], such as the time of day and location [8], but for fast charging, this is more ambivalent as variation is low. Analysis of level 2 charging behaviour shows the spatial

heterogeneity is relevant. Although various articles have looked into fast charging patterns [9–11], so far a systematic evaluation of the differences in intentions of EV drivers between fast charging in the city and along corridors is missing. Such information is crucial for an efficient roll-out strategy to be able to make decisions about the speed and dimension of both level 2 and fast charging installations.

## 1.1. Previous Work

It is generally accepted that a vast infrastructure of fast charging stations is needed to facilitate a full transition to electric mobility. As Motoaki [12] states, the question that remains is: *"Where should we put charging stations?"*. So far, most studies used travel patterns from gasoline-driven vehicles and have assumed that charging stations should fulfil the same needs [13–15]. For fast charging stations, it is mainly assumed that the driver would only like to use them if the battery level drops below an uncomfortable level [16], while charging at slower level 2 charging stations is done when the car is parked [17,18]. This has been done to model both inter-city [2,19] and inner-city [20–22] charging needs. Specific charging systems for taxis have been a popular research topic [5,23]. So far, the assumptions behind these models have hardly been checked.

More recently, research has focused on the actual utilization of fast charging stations and the choices of EV drivers for these fast charging stations. Gnann et al. [24] compared fast charging station usage in front-runner countries Norway and Sweden and concluded that the number of fast charging stations does not need to exceed the number of gas stations. Despite the longer charging times compared to gasoline refueling, there is no need for more charging stations due to home and workplace charging being the dominant charging modes. These findings are supported by a large study on the vehicle level in the United States. This study showed that only 8% of charging is done at DC fast charging stations [25]. Analysis of choices of Battery Electric Vehicle (BEV) drivers in Japan for fast charging stations revealed that drivers are willing to detour up to 1750 m to find a station [26]. Free charging and nearby services of a gas station increased the likeliness that a fast charging station was chosen. Most fast charging was done with a low state-of-charge (SoC) of the battery. Motoaki and Shirk [9] showed that installing a flat fee for fast charging is not efficient in terms of utilization as EV drivers want to get the most value for their money. At fast charging stations, the charging speed declines with higher SoC. These studies provide a first insight into preferences for fast charging stations. Developments of EVs in battery size and charging speeds are not yet taken into account.

## 1.2. Contribution

Previous work has already focused on the charging behavior of electric drivers at level 2 and fast charging stations. This paper focusses on fast charging stations in densely populated areas, with a specific focus on European cities in which many residents rely on on-street parking and charging. While many cities have focused on a roll-out of slower level 2 charging as a substitute for lacking home and office charging possibilities, this research looks at the possibilities of using inner-city fast charging stations to fulfil charging needs. To understand how such a fast charging system should look like to fulfil EV drivers' needs, this work uses three different methods.

First, this study sheds light on the question on whether the idea behind the difference in recharging needs at level and fast charging are true. To this end, level 2 and fast charging infrastructure charging patterns are compared using charging transaction data (>1 million charging sessions) of public charging infrastructure in the city of Amsterdam. The results are compared to previous work. Secondly, a survey is done amongst users of fast charging stations to gain more in-depth insight into the motivations of EV drivers to use fast charging stations. The survey also provides insight into the extent fast charging station usage is an option for those that do not have home charging available. Additionally, it provides new ideas on how different types of modalities, such as taxi drivers, differ from regular users. Thirdly, this paper presents the results of a stated choice experiment for charging station choice. Using a scenario in which the EV driver relies on on-street charging (as 70% of Dutch drivers do), the choice experiment allows us to investigate under which circumstances fast charging can be a

substitute for level 2 charging. Varying, amongst others, the charging speed of the station, it gives further insight for policy makers and charging point operators on how to plan further roll-out in future scenarios. The study, therefore, provides more information on fast charging station deployment in future scenarios. The combination of these three methods allows policy makers to gain further insight into the current utilization of and motivation behind fast charging and how this is likely to develop into the future given current preferences.

## 2. Methodology

This study relies on three types of data collection. First, actual usage of fast chargers within the four major cities within The Netherlands are compared with level 2 charging stations. Data are analyzed and comparisons are made between the two types of charging stations. Secondly, a survey is carried out to collect more in-depth information about the user's motive to charge at a particular fast charging station. Thirdly, a stated choice experiment on fast charging is conducted. The methodology for each data collection is explained below.

### 2.1. Charging Data

First, usage patterns within the city of fast chargers and level 2 chargers are compared to illustrate the differences and similarities between these types of charging modes. This analysis is based upon a large dataset of 1.4 million charging sessions in four cities in The Netherlands, see [8,27] for a description of the dataset. Data are collected between 2016 and 2018. Of those sessions, 52.190 (3% of total) are collected at seven (0.5%) different fast charging stations. Comparisons are performed on the timing of the sessions, volume charged, and the time connected. A *t*-test is used to indicate statistical differences between these types of charging sessions.

### 2.2. Survey

A survey among fast charge station users at five different, both corridor and inner-city, stations is performed to ask drivers about the background and motivation for charging at the particular charging location and type of charger. Fast chargers were qualified as "inner-city" in cases where the fast chargers were located inside urban centers, not adjacent to a highway. Within these criteria, the fast charging stations of Zaandam Amsterdam Haarlem were qualified as "inner-city". The fast charger at the Zaandam location was located next to a convenient store and offered free charging. Fast charging located along highways were qualified as 'highway' fast chargers (alongside the highways A12 and A13). Fast charging stations should have at least a power of 50 kW (both inner-city and highway).

Users are asked about the type of vehicle, state-of-charge, if they could have completed their next trip, duration of the session, their reason to charge, and their type of mode (private, business, taxi, car sharing). Moreover, users are asked why they preferred this charging station over others. In total, data from 100 EV drivers at fast charging stations are collected.

### 2.3. Choice Experiment

After the survey, the same respondents are asked to participate in a choice experiment. Each of the respondents is faced with four scenarios in which the public charging station (not being a fast charger) near their home was occupied or malfunctioning. The choice is between a regular charging station, a charging hub (with 10 connectors), and two different fast charging stations as alternative. The attributes of each the choice options are the distance from home, number of available connectors, charging duration and costs. An example of a choice set is given in Figure 1. The attributes of the regular charging station, the charging hub, and the number of connectors are held constant across the choices. The regular charger is slightly closer than the charging hub (200 versus 250 m) but the charging hub has more connectors (likely leading to more security of accessibility). Respondents are asked to imagine if they had no information about whether or not the chosen charging station was occupied.

| | Regular | Charging Hub | Fast charger A | Fast Charger B |
|---|---|---|---|---|
| **Number of connectors** | 2 | 10 | 2 | 2 |
| **Distance to home** | 200 meters | 250 meters | 150 meters | 700 meters |
| **Charging Time** | 4 hours | 4 hours | 20 minutes | 5 minutes |
| | | | | |
| **Price** | €0.30/kWh | €0.30/kWh | €0.00/kWh | €0.60/kWh |

Choice     ☐     ☐     ☐     ☐

**Figure 1.** Exemplary choice set.

For the fast charging stations, the distance to home, the charging time, and the price is varied. An overview of the attribute levels is given in Table 1. Attribute levels are determined by looking at currently available levels for the distance and price and often mentioned fast charging speeds in the future. The charging time is based upon a session in which 24 kW needs to be charged. A four-level-three-attribute Taguchi orthogonal design [28] is used. A reversed order is used for fast charger B. Dominant options are altered. The complete choice set is presented in Appendix A and the correlation table in Appendix B. In total, 16 choice options are derived, which are blocked in 4 different sets.

**Table 1.** Choice experiment parameters and levels.

| Attribute | Levels |
|---|---|
| Distance to home | 150 m<br>400 m<br>550 m<br>700 m |
| Charging Time | 5 min<br>10 min<br>15 min<br>20 min |
| Price | €0.00/kWh<br>€0.20/kWh<br>€0.40/kWh<br>€0.60/kWh |

To account for heterogeneity in the preferences across the respondents, a mixed logit model is used for analyzing the data [29]. Models are estimated using the Bison Biogeme software package [30]. Constants for the slower level 2 and charging hub options are estimated, using the fast charging as reference level. The utility function for the charging hub contains both a constant for slower charging as one for a charging hub. Models are tested with 125 to 1000 Halton draws; model estimates remain constant with an increasing number of draws. The final model is drawn with 800 Halton draws.

## 3. Results

### *3.1. Charging Data*

#### 3.1.1. Time of Day

Figures 2 and 3 show the distribution of the start times of charging sessions at fast and slow (level 2) charging stations, respectively. The graphical representation shows that the timing of the charging sessions at these speeds is significantly different. Charging sessions at fast chargers have a

distribution that is mainly concentrated around the center of the day. During night time, far fewer charging sessions are started.

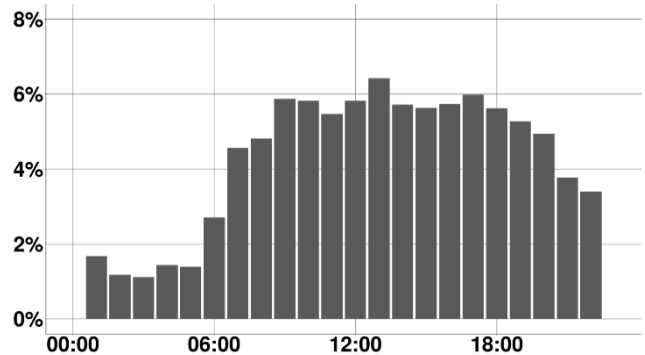

**Figure 2.** Distribution of charging sessions for fast chargers over the day.

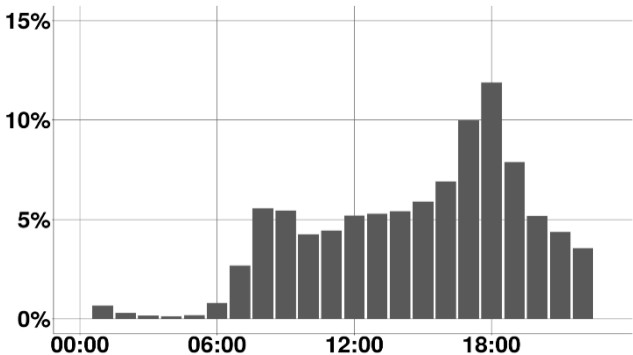

**Figure 3.** Distribution of charging sessions for level 2 chargers over the day.

Charging sessions at level 2 charging stations have a completely different pattern. They show a small peak in the morning and a larger peak in the afternoon. This pattern can be typically observed because many of these charging stations were placed on-demand [31]. These charging stations are mostly used by those that rely on on-street parking for home or office charging. The patterns clearly indicate that slow and fast charging stations are used at different times and, therefore, also for different purposes.

### 3.1.2. Charging Volume

Figures 4 and 5 show the distribution of the volume (kWh) charging at sessions at fast and slow (level 2) charging stations, respectively. The distribution at the fast charging stations is uniform and declines only after 20 kWh. These charge volumes are likely sufficient for drivers to continue their trip. Hardly any charging sessions have more than 30 kWh charged, most likely because trips do not last much longer than another 100 km. The average kWh charged at fast charging stations (10.2 kWh) and level 2 charging station (8.3 kWh) differ significantly from each other (t (58,844) = 62.813, $p$ = 0.00).

The distribution of charging volumes at level 2 charging station shows that smaller volumes are charged at these stations. Most striking is that the number of charging sessions with more than 10 kWh charged is very limited (in total, less than 20% of all sessions; versus more than half of the sessions at fast chargers). The distribution is likely to be different due to a higher share of plug-in hybrid electric vehicles (PHEVs) charging at these charging stations. PHEVs cannot charge at fast chargers, nor do they have the urgency for intermediate charging given the combustion engine as backup. For full electric vehicles, we see a comparable number of kWh charged at both level 2 and fast charging stations. An exception are vehicles with large battery packs (e.g., >60 kWh). These vehicles charge significantly more per session on level 2 chargers compared to fast chargers. This could have two reasons. First,

completely charging the battery at a fast charging station takes more than an hour, which is a long time to stop. Second, many of the cars with large battery packs are from Tesla, which has its own dedicated fast charging network. These data are not included in the analysis.

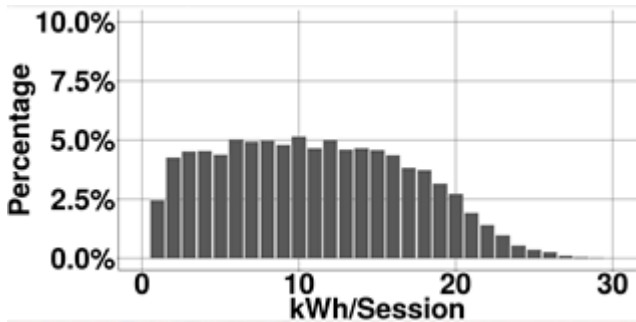

**Figure 4.** Distribution of charging volume at fast charging stations.

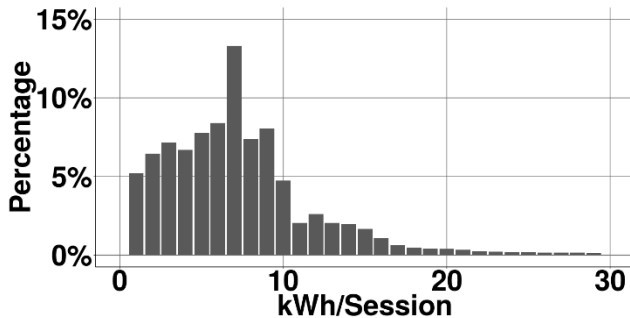

**Figure 5.** Distribution of kWh charged at level 2 charging stations.

### 3.1.3. Connection Time

As expected, the connection times at fast charging station are relatively short. Most charging sessions stop after half an hour and there are hardly any charging sessions longer than 1 h (Figure 6). The need for longer charging sessions does not seem to be present. This is a strong indication that fast charging is the necessity charging, cases in which the battery has run empty, as indicated in previous studies [16,32]. The average connection time at fast charging stations (0.59 h) and level 2 charging station (11.09 h) differ significantly from each other (t (62,837) = −151.14, $p = 0.00$).

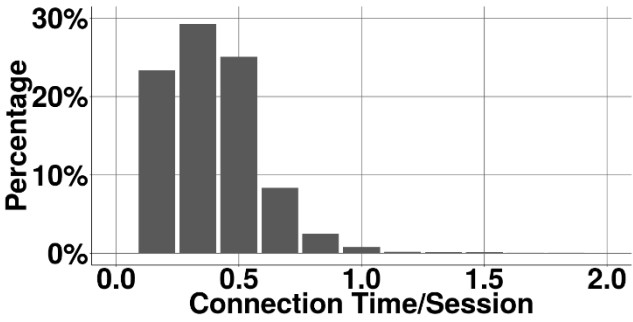

**Figure 6.** Distribution of connection times at fast charging stations per 10 min.

This difference is not surprising as the charging speed at level 2 stations (Figure 7) is much lower (3.7–11 kW). Charging sessions often last much longer than necessary to fully charge the battery. These charging stations have been placed at parking locations near people's homes, work, or points of interest. Therefore, many of these charging stations are also used as a parking spot. They are used overnight or even for entire weekends [33], resulting in much longer connection times than the

difference in charging speed would suggest. These differences indicate very different motivations for using fast and level 2 charging stations.

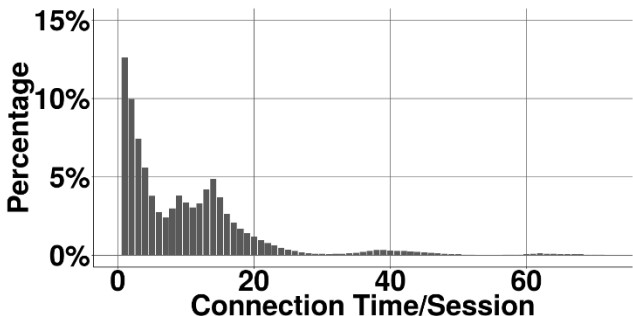

**Figure 7.** Distribution of connection times at level 2 charging stations binned per hour.

### 3.1.4. Conclusion

From the charging data analysis, it can be concluded that fast charging stations and level 2 charging stations are used in a completely different manner. The data suggest that fast charging stations are used to fill up the electric vehicle when necessary. These charging sessions are more likely during trips than at a destination, and vehicles are not likely to be fully charged. Level 2 stations are used either for opportunity charging or to charge at the office or at home while parking. This results in different occupation of these stations across the day and differences in the expected connection time at the station. The amount charged at the stations also differs as plug-in hybrid vehicles only make use of level 2 charging stations. Moreover, full electric vehicles with large battery packs are more likely to completely fill the battery at level 2 charging stations; and not at fast chargers. These results are in line with expectations and findings in previous research [34,35]. Although charging behavior at fast charging stations suggest that it is used for topping up, so far, evidence for motivations for fast charging is lacking.

### 3.2. Survey

The data analysis shows that level 2 and fast charging stations are used for different purposes. For level 2 charging stations, the motivations behind charging can be derived from looking at the charging patterns. Charging patterns at fast charging stations are more uniform. The survey helps to better understand the motivations behind using fast charging stations.

### 3.2.1. General Information

Table 2 provides an overview of the data collected at the charging stations. The average charging time during the survey was found to be 23 min, significantly lower than the 39 min found in the data collection in Section 3.1. The average acquired range is 87 km, corresponding to approximately 16 kWh (assuming 0.2 kWh/km). This is much higher than the average 10 kWh in Section 3.1. Such differences seem surprising as the offered charging speed is equal at all of the locations (50 kW). Yet, charging speeds mainly depend on vehicle restrictions; the stock of EVs that participated in the survey are likely to be different than the general population. The average time at the next destination was nearly 6 h, although the variance is high.

**Table 2.** General information of survey results.

| Charging Characteristic | Average | Minimum | Maximum |
|---|---|---|---|
| Charging time | 23.5 min | 10 min | 66 min |
| Range at start | 91 km | 28 km | 180 km |
| Range acquired | 87 km | 16 km | 148 km |
| KM until next destination | 29 km | 1 km | 170 km |
| Time spend at next destination | 5.9 h | 0 h | 115 h |

### 3.2.2. Trip Purpose

Table 3 provides an overview of the trip purpose the drivers are undertaking at the moment of fast charging. Striking is the high percentage (47%) of taxi drivers that participated in the survey, which reflects the high utilization of fast charging stations by this group. This group nearly always charges (85% of the surveyed taxi drivers) at the inner-city chargers. Many of the taxi drivers at this location operate their taxis at the Schiphol airport in Amsterdam. They do not, however, have a fast charging station at this location, leading them to drive to the nearest inner-city charger after driving a person downtown. Other trips are mainly related to work in some way, such as the 27% of business trips and the 13% of commute charging.

**Table 3.** Distribution of trip purpose.

| Type of Trip | Share |
|---|---|
| Private | 11% |
| Business | 27% |
| Commute | 13% |
| Taxi | 47% |
| Other | 2% |

### 3.2.3. Main Reason to Charge

Based on the survey, the main reason to use a fast charging station is Time left and possibility to charge (Table 4). This is surprising as researchers so far assume that fast charging is only used in case a vehicle has insufficient range. Only 17% of the drivers indicate that this was the reason to use this particular station. More surprisingly, at the corridor locations, the response was the same as at inner-city chargers. Also, at this location, most people answered that they had time left anyway. Other reasons include the fast charging speed, having no charging station at home, and it being cheaper than regular charging. The latter mainly applies to respondents that are surveyed at the free charging station in Zaandam.

**Table 4.** Main reason to charge.

| Main Reason to Charge | Share |
|---|---|
| Insufficient range | 17% |
| No charging station at home | 4% |
| Cheap | 8% |
| Charging speed | 15% |
| Time left and possibility to charge | 52% |
| Other | 4% |

### 3.2.4. Possibility to Charge at Next Destination

Drivers are also asked if they would be able to charge at their next destination, which could provide more insight into the reasons to use a fast charging station. Despite the rather dense charging infrastructure in The Netherlands, only 29% percent can charge at their next destination. Even for only those that stay at least two hours at the next location, the share of having a charging opportunity remains similar. A large portion of the people respond that they do not know (27%) if they could charge. The large share of taxi drivers is a part of the explanation. Of the taxi drivers, only 3% respond affirmative to being able to charge at the next destination. About a third (29%) of the respondents do have the possibility to charge at the next location. All in all, the majority of fast charger users do not have the (knowledge of) the opportunity to charge at their next destination.

### 3.2.5. Conclusion

The survey reveals that there is a large variety in trip purpose when making use of fast charging stations. Taxi drivers make intensive of use of fast charging stations especially at inner-cities stations along routes between often visited locations such as between the airport and the city. Surprisingly, fast charging is not only done to refill the battery when empty, but drivers often used charging stations when they have time left anyway. Fast charging is hardly (4%) a substitute for home charging, but many of the drivers do indicate that no charging station is available at their next destination or this is unknown. This implies that a share fast charging demand is induced from a lack of destination chargers.

### 3.3. Choice Experiment

The same 100 respondents that filled in the survey also participated in the choice experiment. In total, 397 useful choices are made. In Table 5, the results of the mixed logit model are presented. Interactions with different variables, such as if the vehicle is owned by a taxi driver, were tested but a simple model with all variables modelled as continuous provides the best model fit. A multinomial logit model was estimated as well but provided a significant lesser fit (Final LL = −406.66). The mixed logit model provides a good fit to the data (adjusted rho-squared: 0.351).

**Table 5.** Results of mixed logit model.

| Estimates | Value | Rob. Std err. | Rob. $t$-Value | $p$-Value |
|---|---|---|---|---|
| Constant slow | −2.95 | 0.973 | −3.03 | 0.00 |
| Constant hub | −3.99 | 0.666 | −5.99 | 0.00 |
| Fast Charging price | −0.0641 | 0.0086 | −7.45 | 0.00 |
| Fast Charging time | −0.0437 | 0.0165 | −2.66 | 0.02 |
| Fast Charging Distance | 0.000175 | 0.000575 | 0.30 | 0.75 |
| Sigma slow | 3.18 | 0.909 | 3.50 | 0.00 |
| Sigma hub | 2.70 | 0.452 | 5.96 | 0.00 |
| Number of observations | 397 | | | |
| Number of individuals | 100 | | | |
| Null log likelihood | −550.359 | | | |
| Final log likelihood | −349.725 | | | |
| $\rho^2$ | 0.365 | | | |
| *Adjusted $\rho^2$* | 0.352 | | | |

The negative constants for slower charging and the charging hub show that EV drivers in general have a positive attitude towards fast charging as an alternative for home charging when not available. These results are quite remarkable as, in general, most EV drivers make use of slower charging facilities

for most of their charging. It is important to note that fast chargers are more preferred to slower charging options that are at least 200 m away. Closer level 2 charging alternatives could well be preferred over fast charging. During the experiment, the distance to the regular charger as the charging hub was held constant as the focus was on the fast charging parameters. Other research has found that walking distance to the assumed destination is a relevant parameter [36,37]. Additional evidence for this argument is found in the fact that the constant for the charging hub is negative, as in this experiment the hub was fixed at a larger distance from the home location. The certainty of having multiple connections available at the hub was not highly regarded by the respondents.

The parameters for charging price and charging time at fast charging stations are negative and significant. The distance parameter, however, is not significant, which is in line with expectations; a few hundred meters of driving takes little time compared to the charging time. The price and charging time parameters suggest that in the case of fast charging at 50 kW (the current standard) and with double the price of regular charging (€0.60/kWh), the majority of EV drivers still prefer the slower option. In case of ultra-fast charging (350 kW) in combination with a similar price to regular charging (€0.30/kWh), the majority of EV drivers prefer the fast charging station. These results indicate that ultra-fast charging can be a substitute for regular charging when such charging stations are not in the proximity of the intended charging location.

## 4. Conclusions

This paper uses three different data sources on assessing charging behavior on and motivations for using fast chargers. The first analysis on actual charging data from the fast chargers showed that they are used significantly different than level 2 chargers on all aspects analyzed. These include the timing of the session, the volume charged, and the length of the charging connection. This analysis has provided additional evidence that level 2 and fast charging stations cannot be used interchangeably in the planning of charging infrastructure. Especially in dense urban areas, such as The Netherlands, level 2 charging stations are better in serving so-called 'park-and-charge' behavior (or destination charging). This includes overnight and office charging for those that rely on on-street charging. Fast charging serves the needs for those that require a quick fill-up (opportunity or urgency charging).

The second data source is a survey performed among those that were fast charging at both inner-city and corridor fast chargers provided insight into what the reason for using a fast charging station. Surprisingly, this survey finds that the majority of charging sessions is not due to insufficient range to complete the trip, but the main reason was time left and the possibility to charge. Even at the corridor locations, this was the main reason to charge. The amount of data collected was limited, lacking the opportunity to compare, for example, the time of day and weekdays and weekends at each location. Further research could try to expand on such factors. The high number of taxis at the fast charging stations gives an indication that fast charging infrastructure for different modes of transport could be required. Fast charging for taxis could be placed at typical lunch stops or when they have longer dwell times for clients. In future research, a comparison between the results for taxis and different modes, such as car sharing or city logistics, can be made. Different modes of charging could be required for modes and locations.

Lastly, the results provide evidence for the idea fast charging can be a substitute for public home and workplace charging in case the EV driver does not have a private parking spot available and a slow charging alternative is not in their proximity. The fast charging speed should, however, be much higher than the current standard. In such a situation, the additional time of having to walk from a slower charging station to the destination would be comparable to the fast charging time. Prices for fast charging should also be comparable to level 2 alternatives for fast charging to become a preferred alternative. Other factors that have not been taken into account such as possible waiting times and nearby facilities may also play a significant role, especially when fast charging speeds are still limited. An approach in which fast charging stations can be a serious alternative or provide substantial relief to an inner-city level 2 charging network is multi-faceted. Such fast charging stations should

provide sufficient charging speeds, comparable prices, and additional services to make waiting times more enjoyable.

The results indicate that the differences found in current charging behavior at fast and level 2 charging stations are mainly due to the technological restrictions of EV charging stations. The 'slower' fast charging speed, at least compared to gasoline refueling, and higher prices cause EV drivers prefer home and destination charging over fast charging. Technology developments should happen both at the charger as well as the battery side of the vehicle. Battery technology should improve to allow fast charger rates. As well, larger battery capacities should result in less need for daily charging, increasing the acceptance of occasional necessity charging at fast charging stations. Technological developments shift preferences to fast charging if a good business case for fast chargers can be developed in which prices can compete with regular charging. Earlier research on fast charging business cases [38], however, indicate that such opportunities can be very different across countries. Fast charging at speeds such at 350 kw is still years away due to technological restrictions, mainly on the battery side. Policy makers should be aware of possible lock-in effects for level 2 charging as this is likely to remain the favorable option for the years ahead.

**Author Contributions:** Conceptualization, R.W.; investigation, R.W.; methodology, R.W.; visualization, R.W.; writing—original draft, R.W. and R.v.d.H.; writing—review and editing, R.v.d.H.

**Acknowledgments:** We are grateful for the funding provided by the Sia Raak for the IDOLaad project of which this research is part of. We are also grateful for the cooperation of the participating municipalities and charging station operators for providing the relevant data.

**Conflicts of Interest:** The authors declare no conflict of interest.

## Appendix A. Choice Set Design

**Table A1.** Parameters for fast charging stations are displayed. The parameters for the slower charging stations were kept constant.

| Price A (/kWh) | Distance A (meters) | Charging Time A (min) | Price B (/kWh) | Distance B (meters) | Charging Time B (min) |
|---|---|---|---|---|---|
| €0.00 | 150 | 20 | €0.60 | 700 | 5 |
| €0.20 | 700 | 15 | €0.60 | 550 | 15 |
| €0.40 | 400 | 20 | €0.60 | 700 | 5 |
| €0.60 | 700 | 5 | €0.00 | 550 | 15 |
| €0.00 | 400 | 10 | €0.40 | 150 | 15 |
| €0.20 | 550 | 20 | €0.20 | 700 | 15 |
| €0.40 | 150 | 15 | €0.40 | 700 | 10 |
| €0.60 | 550 | 10 | €0.00 | 700 | 20 |
| €0.00 | 550 | 15 | €0.60 | 150 | 20 |
| €0.20 | 400 | 5 | €0.20 | 150 | 10 |
| €0.40 | 550 | 5 | €0.60 | 400 | 15 |
| €0.60 | 400 | 15 | €0.40 | 400 | 20 |
| €0.00 | 700 | 20 | €0.20 | 400 | 5 |
| €0.20 | 150 | 10 | €0.00 | 400 | 10 |
| €0.40 | 700 | 10 | €0.20 | 550 | 20 |
| €0.60 | 150 | 20 | €0.40 | 550 | 5 |

## Appendix B. Correlation Matrix

**Table A2.** Correlation Matrix.

|  | Price A | Distance A | Charging Time A | Price B | Distance B | Charging Time B |
|---|---|---|---|---|---|---|
| **Price A** | 1 |  |  |  |  |  |
| **Distance A** | -0.21439 | 1 |  |  |  |  |
| **Charging time A** | -0.02805 | −0.07817 | 1 |  |  |  |
| **Price B** | 0.020615 | 0.02394 | 0.200452 | 1 |  |  |
| **Distance B** | 0.292966 | −0.16495 | 0.612889 | 0.171304 | 1 |  |
| **Charging time B** | 0.0799 | 0.453617 | −0.43161 | 0.028551 | −0.36885 | 1 |

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
