# Peer review of "Electric Vehicle Fast Charging Needs in Cities and along Corridors"

_wevj, doi:10.3390/wevj10020045_

Round 1

Reviewer 1 Report

Overall, the authors present an Interesting study for comparing the preferences among ev drivers toward their main recharging type. However, I find the results a bit obvious in the sense that home and level 2 charging levels are not supposed to be replaced by level 3 or fast charging entirely, on the contrary, they should complement each other to provide alternatives for EV drivers and allow to use their cars as conventional ones. This issue makes the comprehension and understanding of the paper hard to follow, especially because the conclusions obtained are the same as several authors, in particular the analysis provided in the following reference:

- D. Bowermaster, M. Alexander and M. Duvall, "The Need for Charging: Evaluating utility infrastructures for electric vehicles while providing customer support," in IEEE Electrification Magazine, vol. 5, no. 1, pp. 59-67, March 2017.

Having say this, is a bit hard to quantify the contribution of the paper, I think the authors should reformulate the motivation behind the work, in order to properly understand its value. Comparing the charging patterns of charging methods that are conceptually so different is not useful in terms of usage, charging times, energy charged, etc if we are thinking that one will replace the other one. I think this data can be useful to fully understand the behavior of the average EV driver and understand that depending on each case, a driver will be more inclined to use fast charging and others will rely on home charging, and this difference will be evident in densely populated areas where not everyone has a parking spot on their homes or a nearby public charger.

- Another suggestion is that these driving behaviors are only representative of European EV drivers, given that their driving regime is completely different from North American ones (a bigger rely on public transportation, reduced travel distances, a strong train network for inter country travels, etc). Because of these differences, the study should emphasize the relevance of the results to European countries only. It may not be clear for the readers

- There is a common mistake when comparing fast charging processes and it appears at the end of section 3.2.1, despite several cars are charged with the same fast charger, the actual power level during the charging process will depend on the Battery Management System and most likely in the manufacturer and model of the car, besides the state of charge, state of health and many other variables. The 50 kW is simply the rated power of the charging port, showing how much is it maximum power. Thereby, the differences obtained in the study are not that surprising, given that there are several models/manufacturers that were charged by the mentioned charging stations.

Typos:

- line 125 says: numer of connectors

- line 135, do you mean that the charging times are based on a 24 kWh base? I assume that you were trying to standardize the capacity of the battery in order to establish a fair comparison.

- line 135, I suggest using: four-level-three-attribute Taguchi orthogonal design.

- line 210: This sentence is confusing, do you mean?: "The data shows that these (fast charging stations) HAVE(?) more charging sessions during trips than at a destination and vehicles are not likely to be fully charged".

- line 211: Please rephrase:  Level 2 stations are USED either USED for opportunity charging.

Author Response

Please find comments in attached file

Reviewer 2 Report

The authors do present a paper on a very well-known topic such as the Electric vehicle fast charging needs based on three different collective data and results. Despite the small amount of users (100) used from the survey data which could be statistically speaking not so representative to extrapolate these results for all the existing charging stations and users, still the output in combination with the rest of data analysis can show the importance of this analysis which at the end would be able to benefit towards the right use of the fast charging stations as a function of the users behavior. For those reasons the paper is suggested to be accepted as it is.  

Author Response

Please find comments in attached file
